# Plasmalogens Improve Lymphatic Clearance of Amyloid Beta from Mouse Brain and Cognitive Functions

**DOI:** 10.3390/ijms252312552

**Published:** 2024-11-22

**Authors:** Alexander Shirokov, Daria Zlatogosrkaya, Viktoria Adushkina, Elena Vodovozova, Kristina Kardashevskaya, Ruslan Sultanov, Sergey Kasyanov, Inna Blokhina, Andrey Terskov, Maria Tzoy, Arina Evsyukova, Alexander Dubrovsky, Matvey Tuzhilkin, Inna Elezarova, Alexander Dmitrenko, Maria Manzhaeva, Valeria Krupnova, Anastasiia Semiachkina-Glushkovskaia, Egor Ilyukov, Dmitry Myagkov, Dmitry Tuktarov, Sergey Popov, Tymophey Inozemzev, Nikita Navolokin, Ivan Fedosov, Oxana Semyachkina-Glushkovskaya

**Affiliations:** 1Institute of Biochemistry and Physiology of Plants and Microorganisms, Russian Academy of Sciences, Prospekt Entuziastov 13, 410049 Saratov, Russia; shirokov_a@ibppm.ru; 2Department of Biology, Saratov State University, Astrakhanskaya Str. 83, 410012 Saratov, Russia; eloveda@mail.ru (D.Z.); adushkina.info@mail.ru (V.A.); inna-474@yandex.ru (I.B.); terskow.andrey@gmail.com (A.T.); arina-evsyukova@mail.ru (A.E.); tuzhilkinma@yandex.ru (M.T.); inna.elizarowa7@yandex.ru (I.E.); admitrenko2001@mail.ru (A.D.); mariamang1412@gmail.com (M.M.); krupnova_0110@mail.ru (V.K.); nastya.glushkovskaya04@mail.ru (A.S.-G.); nik-navolokin@yandex.ru (N.N.); 3Shemyakin–Ovchinnikov Institute of Bioorganic Chemistry, Russian Academy of Sciences, Ul. Miklukho-Maklaya, 16/10, 117997 Moscow, Russia; elvod.ibch@yandex.ru (E.V.); grinovk@gmail.com (K.K.); paskalkamal@mail.ru (A.D.); egor.re01@mail.ru (E.I.); dmyagk0v@yandex.ru (D.M.); ivanov.ivao@yandex.ru (D.T.); fakerdack123@mail.ru (S.P.); inoztim@gmail.com (T.I.); fedosov_optics@mail.ru (I.F.); 4A.V. Zhirmunsky National Scientific Center of Marine Biology, Far Eastern Branch, Russian Academy of Sciences, Palchevskogo Str. 17, 690041 Vladivostok, Russia; sultanovruslan90@yandex.ru (R.S.); serg724@yandex.ru (S.K.); 5Physics Department, Saratov State University, Astrakhanskaya Str. 83, 410012 Saratov, Russia; dethaos@bk.ru; 6Department of Pathological Anatomy, Saratov Medical State University, Bolshaya Kazachaya Str. 112, 410012 Saratov, Russia

**Keywords:** plasmalogens, Alzheimer’s disease, age, cognitive functions, amyloid beta, lymphatic clearance

## Abstract

Amyloid beta (Aβ) is a neuronal metabolic product that plays an important role in maintaining brain homeostasis. Normally, intensive brain Aβ formation is accompanied by its effective lymphatic removal. However, the excessive accumulation of brain Aβ is observed with age and during the development of Alzheimer’s disease (AD) leading to cognitive impairment and memory deficits. There is emerging evidence that plasmalogens (Pls), as one of the key brain lipids, may be beneficial for AD and cognitive aging. Here, we studied the effects of Pls on cognitive functions and the lymphatic clearance of Aβ from the brain of AD mice and mice of different ages. The results showed that Pls effectively reduce brain Aβ levels and facilitate learning in aged but not old mice. In AD mice, Pls improve the lymphatic clearance of Aβ that is accompanied by an increase in general motor activity and an improvement of the emotional status and learning ability. Thus, these findings suggest that Pls could be a promising candidate for the alternative or concomitant therapy of AD and age-related brain diseases to enhance the lymphatic clearance of Aβ from the brain and cognitive functions.

## 1. Introduction

Amyloid beta (Aβ) is notoriously known as a peptide, which is involved in the pathogenesis of Alzheimer’s disease (AD) [1,2,3]. However, Aβ is a normal product of neuronal activity playing a pivotal role in the regulation of brain homeostasis [4,5,6,7]. Indeed, Aβ is a result of the cleavage of amyloid precursor protein (APP), involved in the signal transduction process [8]. APP gives rise to Aβ and thus has a key role in the pathogenesis of AD [9]. However, APP plays an important role in the regulation of the brain physiology, including nervous system development, the formation and function of the neuromuscular junction, synaptogenesis, dendritic complexity and spine density, axonal growth and guidance, synaptic plasticity, and the formation of learning and memory [10]. APP processing leading to Aβ formation or not depends on proteolytic mechanisms, i.e., the amyloidogenic and non-amyloidogenic pathways that differently cleave APP via α-, β-, and γ-secretases and release different proteolytic products [8]. In the amyloidogenic pathway, APP is cut by β- and γ-secretases, leading to the production of different Aβ peptides [8,11]. In contrast, in the non-amyloidogenic pathway, APP is cut by α-secretase in the plasma membrane, cleaving APP within the Aβ sequence and thereby preventing Aβ formation [8,12]. A balanced generation and elimination of Aβ peptide products occurs in the healthy brain [13]. However, in the AD brain, Aβ peptides are more prone to aggregation into toxic amyloid oligomers, that eventually evolve into protofibrils and fibrils, via different assembly processes, affecting neuronal function and synaptic activity, and ultimately leading to synapse loss and an impaired cerebral capillary blood flow [14].

In the normal state, Aβ plays multiple physiological roles, particularly in regulating synaptic transmission and regulating synaptic plasticity [15,16]. Some studies have shown that the deprivation of endogenous Aβ gives rise to synaptic dysfunction and cognitive deficiency, while the moderate elevation of this peptide enhances long term potentiation and leads to neuronal hyperexcitability [15]. Aβ has also neurotrophic, neuroprotective, antioxidant, and antiapoptotic effects [16,17,18,19,20]. Aβ is involved in the regulation of neurogenesis and cholesterol biosynthesis [21]. The synapse is vulnerable to an energy deficiency as a highly energy-consuming structure [22]. It has been shown that subthreshold Aβ deposition is correlated with increased aerobic glycolysis in the human brain [23].

As was established in the culture of neurons, both APP and Aβ are produced at extremely high rates into the extracellular space (up to one molecule per second per neuron) [24]. The abundance of Aβ secretion in brain tissues requires an effective mechanism of its metabolism, degradation, and clearance [25,26]. In a healthy brain, during Aβ metabolism, its soluble forms are rapidly evacuated from the brain through the meningeal lymphatic vessels (MLVs) [27]. However, in an aged brain, due to age-related decline in the MLV functions and the dysregulation of enzymes involved in Aβ degradation, the lymphatic clearance of Aβ is dramatically reduced, leading to an increase in the brain Aβ levels [27,28,29,30,31]. The pathological changes in the brain Aβ levels, predominantly Aβ42, play a crucial role in the development of AD, which is a progressive neurodegenerative disease affecting memory, thinking, and learning in elderly people [31,32,33,34]. Over the past 2 years, AD mortality has increased by 145%, which is associated with the COVID-19-induced stimulation of the excessive accumulation of Aβ in brain tissues and the development of COVID-19-related dementia [35,36,37,38,39].

Thus, Aβ is a product of brain metabolism, performing important functions in maintaining its homeostasis. However, Aβ metabolism changes with age, which is accompanied by a decrease in its elimination from brain tissues, leading to cognitive dysfunction. Therefore, the development of strategies for the improvement of clearance of Aβ from the aging brain may lead to progress in both preventing age-related cognitive impairment and preventing the development of AD in older people. Currently, there are no effective methods for improving Aβ clearance from the brain. New therapeutic strategies are based on passive immunotherapy using monoclonal antibodies against Aβ [40,41,42]. However, two clinical trials (EMERGE and ENGAGE) with the monoclonal antibody Aducanumab demonstrate inconsistent results and negative consequences of anti-Aβ immunotherapy [43,44]. Therefore, to improve this therapy, a combined method has been proposed based on an increase in the clearance of toxic Aβ by stimulating lymphangiogenesis [42]. However, the stimulation of lymphaneogenesis requires the introduction of the vascular endothelial growth factor C (VEGF-C) into the cisterna magna, which is an invasive method that can only be performed by highly qualified specialists which limits its widespread use in clinical practice [42]. Recently, lecanemab (trade name of Leqembi) and donanemab (marketed as Kisunla) are now going through approvals by the Medicines and Healthcare Products Regulatory Agency for the therapy of AD [45].

There is growing evidence that plasmalogens (Pls) may be beneficial for AD and cognitive aging [46,47,48,49,50,51,52,53]. Pls are one of the key brain lipids, which are involved in the regulation of various brain functions [54,55,56]. Pls are a major component of membranes of synapses, synaptic vesicles, and myelin [57,58]. Due to their fusogenic property, Pls play a crucial role in neurotransmission [59]. Pls keep microglial health and are involved in brain immunity [47,60]. Also, Pls affect membrane dynamics, trafficking, thickness, curvature, permeability, and fusion [61,62,63,64], and they facilitate signal transduction processes, including cholesterol efflux [65,66]. Since Pls are found in large quantities in the brain, it is not surprising that their decreased levels are associated with many brain pathologies, including age-related diseases and AD [48,49,50,51]. There are a number of studies suggesting a direct link of Pls deficiency and AD [67,68,69,70].

Recently, human and animal studies have supported the efficacy of Pls for restoring memory, behavioral disturbances, and cognitive dysfunctions [47,71,72]. The oral administration of Pls enhances the synaptic plasticity of the hippocampal neurons and eliminates aging-associated synaptic defects [47]. Pls enhance learning and memory due to the regulation of the brain-derived neurotrophic factor (BDNF) activity [71], which is also the basis of the Pls-mediated inhibition of neuronal apoptosis and rescue from neuronal cell death [56,72,73]. The oral ingestion of Pls improves cognitive function and mental conditions in AD patients [74,75,76]. These data suggest that Pls could be an alternative therapy of age-related and AD-induced cognitive decline, memory deficits, and learning difficulties.

However, studies of the mechanisms of therapeutic effects of Pls on AD are phenomenological in nature and remain poorly understood. It remains unclear what is the primary target for Pls due to the activation of which they enhance or improve cognitive abilities and the removal of Aβ from the brain.

MLVs are important “tunnels” for the removal of Aβ from brain tissues [27]. It is discussed in the scientific community that the development of methods to increase the MLV functions will be an advancement in the treatment of AD [27,28,42]. As noted above, the prospect of lymphoneogenesis activation in the improvement of AD immune-therapy has been shown [42]. Based on these data, it is logical to expect that Pls may have a stimulating effect on the lymphatic excretion of Aβ from the brain, which may be one of the important mechanisms of the therapeutic effects of Pls in relation to AD and the aging brain.

Therefore, in this study, we investigate the effect of Pls on cognitive functions and the lymphatic clearance of soluble Aβ from the brain of male mice using two functional models: (1) the injection model of AD induced by injecting recombinant Aβ 1–42 into the right dentate gyrus of the hippocampus and reflecting the early stages of AD; and (2) the age model depicting natural age-related changes in Aβ accumulation in the brain [77] (Figure 1a–c).

## 2. Results

### 2.1. Effects of Pls on Clearance of Aβ from the Brain and Cognitive Functions: A Model of the Early Stages of AD

In the first step, we answered the question of whether Pls can stimulate Aβ removal from the brain and improve cognitive dysfunction at the early stages of AD. For this purpose, an injection model of AD (unilateral Aβ injection into the hippocampus) was chosen, which allows one to study the early stages of Aβ-induced changes in brain tissues and functions [78,79,80]. The studies were conducted in four groups: (1) the control (3-month-old mice); (2) the control + Pls; (3) AD (3-month-old mice); and (4) AD + Pls. Figure 2a,e,i,m–o show that in the control, the small amounts of Aβ are present in brain tissues, its meninges, and in dcLNs, which is consistent with the results of other researchers [77,81]. At the same time, Pls did not change the Aβ level in the studied tissues in the control (Figure 2b,f,j,m–o). In the AD group, the Aβ level was significantly higher than in the control group in all studied tissues (Figure 2c,g,k,m–o). It is important to note that a 21-day course of Pls reduced the level of Aβ in the brain, its meninges, and in dcLNs in AD mice (Figure 2d,h,l,m–o). The ELISA analysis confirmed the confocal data and demonstrated the increased level of Aβ in brain tissues of mice from the AD group, which decreased after the Pls course (Table 1). There was no effect of Pls on the Aβ amount in the brain in the control groups, possibly due to the low level of brain Aβ in the normal state (Table 1).

Thus, these data indicate that the increased content of Aβ in the brain and lymphatic system of AD mice can be improved by a 21-day course of Pls.

Compared to the control, AD mice showed no cognitive impairment in spatially demanding tasks, i.e., performing the Y-maze test and the new object recognition test (Figure 3h,k). However, AD mice demonstrated lower motor activity and higher anxiety compared to the controls, as measured by the total distance traveled and time spent in the center of the arena (Figure 3b,c). Note that these behavior parameters were better in AD mice after a course of Pls, i.e., Pls improved the motor activity and emotional state of AD mice (Figure 3b,c). The most pronounced effects of Pls were observed in the study of the higher nervous activity of mice. Indeed, the development of Pavlov’s conditioned reflex (light-food) was significantly hampered in AD mice compared to the control (Figure 4d,e). This was expressed both in a larger number of sessions required for the stable formation of this reflex and in the lower number of rewards in the final session (Figure 4d,e). A 21-day course of Pls significantly facilitated the ability of AD mice to develop a conditioned reflex. So, under a course of Pls, AD mice demonstrated both a higher rate of development of the conditioned reflex “light-food” and a higher number of unconditioned rewards (food) for the presentation of a conditioned signal (light) (Figure 4d,e).

These data indicate that the early stagy of AD is characterized by a decrease in general motor activity and an increase in anxiety, which is accompanied by a reduced ability to form new conditioned reflexes. The early signs of AD can be improved by Pls, which increases the motor behavior, emotional status, and learning function.

### 2.2. Effects of Pls on Clearance of Aβ from the Brain and Cognitive Functions: An Age Model of Aβ Plague Formation

The increased Aβ levels in brain tissues are not only one of the signs of AD development, but also a factor that accompanies brain aging [64,78,82]. To answer the question of whether it is possible to correct age-related changes in the brain related to Aβ levels and cognitive functions, the second series of studies examined the effects of a Pls course on Aβ removal from the brain into dcLNs as well as on performing the behavior tests in mice at different ages—3-month-old (young adult), 6-month-old (middle age), 14-month-old (aged), and 24-month-old (old) mice.

The results of the confocal analysis revealed an increased Aβ level in the brain and in the meninges in aged mice and, especially, in old mice compared with young and middle-aged animals, which demonstrated a similar low Aβ level in the tested tissues (Figure 5a–h,m,n). Interestingly, in 24-month-old mice, despite the high Aβ values in the brain and in the meninges, there was a low Aβ content in dcLNs (Figure 5l,o), which is possibly due to an age-related decrease in the drainage function of MLVs [27,28]. The quantitative ELISA analysis also showed an increase in the brain Aβ level in aged and especially in old mice (Table 1).

The confocal data show that Pls improved Aβ clearance only in 14-month-old mice. Indeed, a 21-day course of Pls reduced the Aβ level in the brain, its meninges, and in the dcLNs in aged mice (Figure 5c,g,k,m–o). The ELISA analysis of the soluble Aβ level in the brain showed similar data to the confocal results (Table 1). However, there were no Pls effects on the removal of Aβ from the brain and the meninges in other age groups (Figure 5a,b,d–f,h–j,l–o). The ELISA findings also revealed that the level of Aβ in the brain was not affected by Pls in 3-, 6-, and 24-month-old mice (Table 1). The lack of effects of Pls on Aβ clearance in young and middle-aged mice may be due to its low content in the brain, while in old mice, it might be due to the age-related dysfunction of MLVs [27,28], which is an important pathway for Aβ clearance from brain tissues to the peripheral lymphatics [27].

Thus, Pls improves Aβ clearance in aged, but not in old mice, most likely due to the partial preservation of the MLV functions in the aged brain [83]. It can explain the low Aβ content in dcLNs in 24-month-old mice, despite a high Aβ level in the brain and the meninges and no Pls effects of Aβ clearance in old mice. In 14-month-old mice, Pls increased the removal of Aβ from the brain to the peripheral lymphatics, i.e., through the lymphatic pathways, leading to a decrease in the level of Aβ in the brain and the meninges and bringing its content closer to that of young and middle-aged animals.

The behavior tests revealed significant age-related differences in locomotor activity and learning, but not in the emotional status (anxiety-like behavior) and short–term memory. So, the open field test, which is extensively used for the study of adaptive behavior to a new environment [84], demonstrated a reduction in the distance traveled, but not in the center time with age (Figure 3d,e). There were no differences between performing the Y-maze and new object recognition tests, suggesting the preservation of spatial and recognition memory in mice of different ages (Figure 3i,l).

The most striking results were obtained in the study of higher nervous activity in mice of different ages. Indeed, the development of the conditioned reflex “light-food” was reduced in aging mice and significantly hampered in old mice compared to young animals (for 14- and 24-month-old mice) and middle-aged rodents (for 24-month-old mice) (Figure 4d,e). There were no differences in performing Pavlovian conditioning between young and middle-aged mice. The course of Pls significantly facilitated the learning of 3-, 6-, and 14-month-old mice, but not of 24-month-old animals. Thus, young, middle-aged, and aging mice taking Pls demonstrated a smaller number of sessions for developing the conditioned reflex “light-food” and a greater number of rewards in the final session than mice without Pls (Figure 4d,e). However, the course of Pls did not affect learning in old mice. These data suggest that stronger changes with age concern higher nervous activity based on a complex learning process and the formation of new synaptic connections. Pls can increase the learning ability only at certain stages of ontogenesis, including young, middle-aged, and aged mice, and have no effect in old animals.

## 3. Discussion

This study examined the effects of Pls on Aβ clearance from a mouse brain to the peripheral lymphatic system and on cognitive functions using two functional models, such as the early stages of AD and age-related changes in the brain. The early stages of AD are characterized by a decrease in the learning ability or the development of the conditioned reflex, but not performing spatially demanding tasks, i.e., the Y-maze test and the new object recognition test. Our data are consistent with the results of other researchers, indicating that not all short-term memory loss is an early sign of AD [85]. One of the hallmarks of early-stage AD is short-term memory loss, i.e., a loss of the ability to perform routine tasks. However, while AD affects memory, it involves far more than simple forgetfulness. A mild memory delay can occur as part of a normal process of brain aging. This explains why traditionally used tests for assessing short-term memory are not effective in the early stages of AD [85]. Indeed, compared with normal animals, AD rats demonstrated no cognitive impairment in performing the Morris water maze and Y-maze tests reflecting spatial memory [86]. The hippocampus plays a crucial role in the formation of spatial memory. There is evidence that the early stages of AD are characterized by a lack of changes in the number of the pyramidal neurons and synaptic destruction that can explain the unaltered spatial short-term memory in early AD [85,86].

However, using the open-field test, we found a decrease in the general motor activity and in an increase in anxiety in AD mice that was associated with a significant reducing ability for learning (the formation of Pavlov’s conditioned reflex “light-food”). Note that a 21-day course of Pls increased the motor behavior, emotional status, and learning in AD animals. Motor activity is an integral part of learning, based on the natural curiosity of mice. Therefore, a Pls-mediated increase in the overall motor activity can also facilitate the formation of a conditioned reflex in AD mice. Learning is a process to transform a neural network to adapt the organism to the new environment. The memory is the state of maintaining such a network and the formation of new synaptic contacts with the modification of their functions, providing a mechanism for learning [87]. There is emerging evidence that Pls could be one of the key phospholipids in the brain, which are critically involved in maintaining cognitive functions and memory [60,88]. Recent studies on postmortem human brain tissues with AD and in the AD model mice show that Pls can improve memory by stimulating the cellular signaling molecules (extracellular signal-regulated kinase and serine/threonine protein kinase) and by activating the membrane-bound G-protein-coupled receptors [71]. It is known that the reduction of brain Pls impairs memory and leads to behavior impairment in both normal and AD brains [48,49,50,51,67,68,69,70,71]. Decreased Pls promotes oxidative damage in AD [89]. The decrease in activity of genes related to the phospholipid biosynthesis, including Pls, significantly reduces long and short memory, while the oral administration of Pls improves memory due to an increase in neuronal cell branching and the number of dendritic spines [71].

Hippocampal neurogenesis is a process in which neural stem cells continue to proliferate and differentiate to produce new neurons [90]. The process of hippocampal neurogenesis is crucial for learning and memory [91,92]. Dysregulated neurogenesis leads to the deterioration of cognitive functions and progressive memory loss, including AD [93]. So, Pls improve neurogenesis in the hippocampus with the upregulation of BDNF-TrkB, signaling that is responsible for the enhancement of memory [75,94]. Li et al. reported that Pls can be a promising candidate for a neurogenesis enhancer and the improvement of the hippocampal-dependent cognitive function of AD mice [88]. Pls therapy protects neural stem cells in the hippocampus from Aβ-induced cytotoxicity via the modulation of the Wnt/β-catenin signaling pathway [88].

In the next step, we studied the Pls effects on Aβ clearance from the brain to dcLNs in the health pf mice of different ages. Our results clearly show age-related increases in the Aβ levels in the brain and its meninges. Indeed, aged and especially old mice demonstrated higher brain and meningeal Aβ levels compared with young and middle-aged mice. Our results are consistent with the findings of normal Aβ formation in the brain as a metabolic product and an age-related increase in the brain level of Aβ due to a decrease in the lymphatic removal of the toxin with age [27,77,81,95]. The results of the Pls therapy confirm this hypothesis. Indeed, Pls improved Aβ clearance in aged, but not in old mice. There is evidence that young and middle-aged mice have no changes in the lymphatic network [27,28]. The process of lymphatic hyperplasia and lymphatic valve dysfunction begins to manifest in aging mice (starting from 13–14 months of age) and progresses to old age, leading to a reduction in brain drainage [28]. In AD mice, Aβ deposition also depends on age-related changes in the lymphatic clearance of the toxin from the brain. So, the deterioration of the lymphatic vasculature at the dorsal MLVs is observed in 13–14-month-old 5xFAD mice, which is accompanied by a significant increase in Aβ deposition throughout the meninges [27,42]. We propose that the preservation of the functional potential of MLVs in aging animals underlies the efficient removal of Aβ from brain tissue to dcLNs. This assumption is based on Da Mesquita’s data, indicating an improvement in Aβ immunotherapy by the stimulation of lymphaneogenesis [42]. The loss of MLVs in the late stages of ontogenesis can explain the low Aβ content in dcLNs in 24-month old mice despite a high Aβ level in the brain and the meninges and no Pls effects of Aβ clearance in them. No effects of Pls were found on the removal of Aβ from the brain in young and middle-aged mice, apparently due to the low Aβ brain level in these age groups. Pls were effective in facilitating learning in all age groups except old mice. Thus, Pls significantly reduced the time of learning Pavlov’s conditioned reflex “light-food” in 3-, 6-, and 14-month-old mice, but not in 24-month-old animals. These data provide an important information basis for a deeper understanding of the use of Pls for the therapy of age-related brain diseases [47,50,96].

*Limitations.* The use of Pls for the treatment of neurodegenerative and age-related brain diseases is a new direction [33,34,35,36,37,38,39,40,47,58]. Therefore, despite promising results [33,36,43,57,59,60], it remains poorly understood what type of Pls is most effective, in what doses, for what duration the therapy should be, in which subjects the Pls therapy is effective and in which it is not, and what method of administration of Pls gives the most pronounced therapeutic effect. In this study, we administered Pls directly into the ventricular system of the brain. This limits the ability to answer the questions of whether the same effects of Pls would be achieved with their use per os or intravenously, and whether Pls pass through the blood–brain barrier. Further research that can answer the above questions will allow us to make significant progress in the development of promising Pls therapy of brain diseases and in better understanding of mechanisms underlying the Pls-related improvement of Aβ clearance and cognitive functions. In our studies, we determined the effects of Pls on the lymphatic removal of soluble forms of Aβ. It seems promising to evaluate the effects of Pls on insoluble Aβ in future studies due to the possible activation of the phagocytic activity of microglia and other immune processes. What might be the best source of Pls also remains to be seen. Here, we used phospholipids from bovine brains since the composition of fatty acids of marine Pls, which have been more studied for oral use, is very different from that of mammals.

It is known that the incidence of AD is higher in elderly women than in men [97,98]. The mechanisms of sex differences in AD resistance remain poorly understood. However, there is evidence that this is most likely not related to the MLV functions, since sex differences in the morphology and activity of MLVs were not found in mature female and male mice [28]. We assume that Pls improve the lymphatic removal of Aβ from the brain equally in both male and female reproductive mice. Nevertheless, our data suggesting a decrease in sensitivity to the Pls therapy with age make it relevant to study sex differences in the effectiveness of Pls in the elderly.

Since our studies were focused on the studying of Pls’ effects on the lymphatic removal of soluble forms of Aβ from the brain, the injection model of AD is most suitable for this purpose. However, the study of the therapeutic effects of Pls on other AD models, including 5xFAD, APP/PS1, 3xTg, etc., is a necessary step in the assessment of the reproducibility of the results.

It should be noted that despite the growing number of scientific works devoted to the study of Pls’ effects on cognitive function improvement in the development of AD, the mechanisms of therapeutic effects remain unexplored. Knowing that Aβ production could be the result of either BACE1 or other APP amyloidogenic processing, it should be expected that a Pls deficiency may upregulate/increase the activity of BACE1 (or other Aβ-producing secretases) rather than γ-secretase. Can the authors explain their line of thought on this matter in a more detailed way? Are Pls reported to modulate BACE1 expression/activity? Moreover, a mechanistic explanation of how Pls can improve AD-related molecular aspects is missing and should be implemented for a comprehensive presentation of the state-of-the-art.

## 4. Materials and Methods

### 4.1. Subjects

Male C57BL/6 mice were used in all experiments and were obtained from the National Laboratory Animal Resource Center in Pushchino (Moscow, Russia). The animals were housed under standard laboratory conditions with access to food and water ad libitum. All experimental procedures were performed in accordance with the “Guide for the Care and Use of Laboratory Animals”, Directive 2010/63/EU on the Protection of Animals Used for Scientific Purposes, and the guidelines from the Ministry of Science and High Education of the Russian Federation (No. 742 from 13 November 1984), which have been approved by the Bioethics Commission of the Saratov State University (Protocol No. 8, 18 April 2023). The mice were housed at 25 ± 2 °C, 55% humidity, and 12:12 h light–dark cycle. The experiments were performed in the following groups: (1) the control (3-month-old, no Pls); (2) the control (3-month-old) + Pls; (3) AD (3-month-old, no PLs); (4) AD (3-month-old) + Pls; (5–7) 6-, 14-, and 24-month-old mice without Pls; (8–10) 6-, 14-, and 24-month-old mice + Pls, n = 7–8 in each group.

### 4.2. Preparation of Liposomes with Pls and Mode of Administration to Mice

Liposomes were prepared using the concentrate of phospholipids from the bovine brain. The concentrate was obtained as described elsewhere excluding column chromatography [99]. Lipid classes were determined by one-dimensional silica gel TLC (thin-layer chromatography). The pre-coated Merck (Darmstadt, Germany) Kieselgel 60 G plates (10 cm × 10 cm) were developed with chloroform/acetone/methanol/acetic acid/water (6:8:2:2:1, *v*/*v*). After drying in a stream of air, plates were sprayed with 10% H_2_SO_4_/MeOH and heated at 180 °C for 10 min. The chromatograms were scanned by an image scanner Epson Perfection 2400 PHOTO (Nagano, Japan) in a grayscale mode. The software 7.0 used for scanning was Adobe Photoshop (Adobe Systems, San Jose, CA, USA). Percentages of lipid contents were determined based on band intensity with the use of the image analysis program Sorbfil TLC Videodensitometer DV (Krasnodar, Russia). The units were calibrated using known standards. The ratio of lipid classes was phosphatidylethanolamines-cerebrosides-phosphatidylcholines-sphingomyelins-phosphatidylserines-sterols, 50:22:20:3.5:3:1.5 (by mol).

To analyze the content and structure of the molecular species including plasmalogen forms of phosphorus-containing lipids, the total lipids were separated on a Shim-Pack diol column (4.6 mm × 50 mm, particle size 5 μm) (Shimadzu, Japan) using a Nexera-e chromatography system (Shimadzu, Japan). Solvent system A (2-propanol–hexane–H_2_O–HCOOH–28% NH_4_OH–Et_3_N, 28:72:1.5:0.1:0.05:0.02, *v*/*v*) and solvent system B (2-propanol–H_2_O–HCOOH–28% NH_4_OH–Et_3_N, 100:1.5:0.1:0.05:0.02, *v*/*v*) were used as eluents. System B content was programmed as follows: 0% (8 min), 0 to 20% (7 min), 20 to 100% (5 min), 100% (15 min), 100 to 0% (0.1 min), and 0% (12 min). The elution rate was 0.2 mL/min. To detect lipids, a high-resolution tandem mass spectrometer LCMS-IT-TOF (Shimadzu, Japan) was used. Analysis was performed under the electro-spray ionization (ESI) mode with simultaneous registration of signals of positive and negative ions. Scanning was performed in a *m*/*z* range of 100–1400. Source voltage was −3.5 kV in case of negative ion formation and 4.5 kV in case of formation of positive ions. The temperature of the ion source was 200 °C; dry gas (N_2_) pressure, 150 kPa; and the flow rate of nebulizing gas (N_2_), 1.5 L/min. Argon (0.003 Pa) was used in the collision chamber of the mass spectrometer. The structural identification of each lipid molecular species was conducted by LC-MS analysis; this entailed comparing the retention times, ion forms, and specific fragmentation behaviors of the phospholipid classes with commercially available lipid standards. The detailed information of identification was described earlier. Percentages of the individual molecular species of each lipid class were calculated by peak area of negative ions [M–H]^−^, except for phosphatidylcholines which were estimated by peak area of negative ions [M+HCOOH]^−^.

The content of docosahexaenoic, adrenic, and arachidonic acids was determined as methyl esters by GC (gas chromatography) and amounted to 7.3%, 5.7%, and 5.1%, respectively. Fatty acid methyl esters (FAME) were obtained by a sequential treatment of the lipids with 1% MeONa/MeOH and 5% HCl/MeOH and purified by preparative TLC development in benzene. GC analysis of FAME was carried out on a Shimadzu GC-2010 chromatograph (Kyoto, Japan) with a flame ionization detector on a SUPELCOWAX 10 (Supelco, Bellefonte, PA, USA) capillary column (25 m × 0.25 mm i.d.) at 210 °C. Helium was used as the carrier gas. FAME were identified by a comparison with authentic standards and using a table of equivalent chain length.

For liposome preparation, the obtained concentrate was dissolved in *tret*-butanol and lyophilized on an Iney-4 (Institute for Biological Instrumentation, Russian Academy of Sciences, Pushchino, Russia) freeze-dryer at 7 Pa. The lipid remnant was hydrated with 10 mM phosphate buffer containing 1 mM ethylenediaminetetraacetic acid and 240 mM sucrose, pH 7.2, under shaking at 37 °C for 2 h. Then, the resulting suspension was subjected to 5–7 cycles of freezing/thawing (liquid nitrogen/+40 °C) and sonicated using a disintegrator UZDZ-0,1/22 (Ultrasonic technologies and equipment Co., Ltd., Saint-Petersburg, Russia) with a tip, 4 times for 1 min at +4 °C with a 1 min interval. Final lipid concentration was 60 mg/mL. Hydrodynamic diameters of the liposomes were measured in diluted dispersions (final lipid concentration 50 µg/mL in the phosphate-buffered saline (PBS)) using a Litesizer 500 (Anton Paar, GmbH, Graz, Austria; semiconductor laser, 658 nm, 90° angle), 3 cycles of 1 min. Effective diameter and polydispersity index were 85.23 nm and 0.226, respectively. The liposome dispersions were divided into aliquots, frozen in liquid nitrogen, and stored at −70 °C on demand. Before usage, an aliquot was thawed and treated for 30 min using ultrasound bath homogenizer Bandelin SONOPULS HD 2070.2, 100 Watt to restore the size of liposomes.

An amount of 5 μL of Pls (~0.15 mg) was injected into the right lateral ventricle (AP-1.0 mm; ML-1.4 mm; DV-3.5 mm) at a rate of 0.1 μL/min using microinjector (Stoelting, St. Luis, MO, USA) with a Hamilton syringe with a 29-G needle (Hamilton Bonaduz AG, Switzerland). The implantation of chronical polyethylene catheter (PE-10, 0.28 mm ID × 0.61 mm OD, Scientific Commodities Inc., Lake Havasu City, AZ, USA) into the right lateral ventricle was preformed according to the protocol reported by Devos et al. [82]. For the study of the Pls’ effects on Aβ clearance from the brain and behavior, a 21-day course of Pls was used. The Pls course was started 7 days after intrahippocampal Aβ injection and/or a catheter implantation into the right lateral ventricle when mice had fully recovered from surgery (Figure 1b).

### 4.3. An Injected Model of Early Stages of AD

To induce AD in mice, we used the injection of Aβ (1–42) peptide into the hippocampus (AP − 2.0 mm; ML +/− 1.3 mm; DV − 1.9 mm). Aβ (1–42) was dissolved in PBS and then incubated for 5–7 days at 37 °C to induce fibril formation [78]. The mice were anesthetized by 1% isoflurane at 1 L/ min N_2_O/O_2_ − 70:30 and fixed in a stereotactic frame. The scalp was removed and the skull surface was dried by clean compressed air. Afterward, Aβ (1 μL, at a final concentration of 1 μg or 0.2 nM) was injected into in the hippocampus at a rate of 0.1μL/min using microinjector (Stoelting, St. Luis, MI, USA) with a Hamilton syringe with a 29-G needle (Hamilton Bonaduz AG, Bonaduz, Switzerland).

### 4.4. Behavioral Testing

The Y-Maze test is used for assessment of short-term working memory in mice, as described previously [100]. The spatial working memory was assessed by allowing mice to explore three arms of apparatus for the maze test (Figure 3f,g). Due to an innate curiosity of mice to explore new areas, mice with intact working memory remember previously visited arms. That is why they avoid entering them and prefer to visit new arms.

To perform this test, we used the Y-maze apparatus with three equal-length arms (which were labeled as A, B, and C) orientated at 120° angles from each other (21 × 7 × 15.5 cm). Mice were placed at a particular position on the Y maze and allowed to explore the apparatus freely for 5 min. Using overhead camera, the number of arm entries and alternations was recorded for 8 min [68]. An entry occurs if all four limbs of mouse were within one investigated arm. An alternation is defined as consecutive entries into all three arms (e.g., in the sequence ACBCABCBCA). The test results were evaluated as the percentage of the alternation behavior, where a high percentage indicates a good working memory.

The movement of rodents is driven primarily by exploratory curiosity [83,101]. The locomotor activity of animals is a manifestation of desire to explore/learn a new space. Therefore, we used the open field test for the assessment of locomotor activity as an additional assessment of cognitive abilities of mice [102]. To perform this test, mice were placed in the center of box (42 × 42 × 42 cm) and their movement was recorded for 10 min using a camera (Figure 3a). The total distance (cm) the mice traveled during the test and the time (sec) spent in the center (25 cm × 25 cm) of the arena were measured.

The new object recognition test was used for assessment of short-term recognition memory [103,104]. Mice were placed in a black box (33 cm × 33 cm × 20 cm) using a video-tracking package (Figure 3j). Two similar objects (red cylinders) were presented to mice for 10 min during the familiarization session. Afterward, during testing session, one of cylinders was replaced by a new object for 10 min (blue cube). Since the quality of test performance is strongly influenced by the experimental environment and external signals [105], the mice were adapted to the test for 10 days. To do this, the experimentalist took them in hands for 1 min every day for 7 days [105]. Furthermore, during habituation phase, mice were placed into the testing arena without any objects for 5 min during 3 days before the test.

We used two asymmetric objects of the same size (cylinders and cube: 3 cm × 3 cm × 3 cm) and odor. Since mice have difficulty in discriminating colors, we selected bright (dark blue and red) objects. The weights of cylinders and cube were heavy enough that the animals could not move them. Patafix held the objects on the surface of floor. The criteria of correct exploration were defined as directing the mouse nose toward the object at a distance of 2 cm or less (Figure 3j). Climbing onto the object or chewing does not qualify as exploration. The results of test were evaluated as the discrimination index by using the formula (T2/(T1 + T2)) × 100, where T1 is the time for exploring the familiar object and T2 is the time for exploring the new object. The discrimination index was expressed in the percentages.

To assess the complex higher nervous activity of mice, we used Pavlovian conditioning. Pavlovian conditioning is a complex test that allows one to evaluate, in natural conditions of a home cage, learning of mice and formation of conditioned connections. The advantages of this test are that in the process of a long time (over the course of a month), mice learn complex actions, which allows one to evaluate the rate of formation of new synaptic connections involved in the development of a conditioned reflex, as well as the interaction of different brain centers involved in the formation of higher nervous activity, including memory, logic, and learning.

We used our modified method of Pavlovian conditioning published in Ref. [106]. To perform this test, an automated operating wall with built-in lever and green light was placed in the home cage (Figure 4a–c). The development of a conditioned reflex according to Pavlov consisted of a hungry mouse being offered the combination “light-food” [107]. Food or reward (12 mg seeds of sunflowers, Grums, Minsk, Belarus) was given through a special (correct) hole, inside which the number of head entries was recorded using an LED (Figure 4a–c). Nearby was a hole through which reward was not given. Mice were placed in an operant chamber for 30 min every day and a camera was used to record the stable formation of a conditioned reflex when the mouse responded to the light and approached the correct hole to receive food. A stable conditioned reflex was considered when the mouse needed 15 s after the light was turned on to reach the correct hole and receive reward. The test included a training session, when the mice were taught to understand that after the light they had to approach the correct hole and receive reward, and a final session, which lasted 30 min with already trained mice. This test assessed (1) the number of sessions (days) required to develop a conditioned reflex, i.e., during the training phase, and (2) the number of head entries in the correct hole and received reward.

In both AD and age models, the behavior tests (the Y-maze, the open filed, and the new object recognition) were performed on 8th day and 28th day of observation (Figure 1b,c). The training session of Pavlov’s conditioned reflex began on 8th day of study and continued throughout a 21-day Pls course. Respectively, the final session of Pavlov’s conditioned reflex was conducted on 28th day of experiments. Since no significant differences in performing spatially demanding tasks and locomotor activity were found in mice of different ages (the Y-maze, the open filed, and the new object recognition), on 28th day of observation, they were tested only for the effectiveness of performing Pavlov’s conditioned reflex.

### 4.5. Immunohistochemical Assay and Confocal Imaging

The Aβ (Aβ1–42) levels were measured in the whole brain, its meninges, and in the deep cervical lymph nodes (dcLNs) using the standard protocols for immunohistochemistry. The choice of these tested tissues was related to the purpose of study of the effects of Pls on the lymphatic removal of soluble Aβ from the central nervous system (CNS), i.e., the removal of Aβ dissolved in the cerebrospinal fluid (CSF), which drains Aβ from brain tissues into its meninges and then into dcLNs, the first anatomical station for collection of CSF with substances dissolved [27,28,29].

The antigen expression was evaluated on sections of the tested tissues according to the standard method of simultaneous combined staining (abcam protocols for free-floating sections) using a confocal laser scanning microscope LEICA TCS SP8 (Leica Microsystems, Wetzlar, Germany). The whole brains were collected and free-floating sections were prepared. Then, 30 min before decapitation, Evans blue dye (1%, Sigma-Aldrich, St Louis, MI, USA) was injected intravenously to fill the cerebral vessels. Afterward, the tested organs were fixed for 48 h in a 4% saline solution-buffered formalin and then were cut on the slices with a thickness of 40–50 microns using a vibrotome Leica VT1000 S (Leica Microsystems, Wetzlar, Germany). The nonspecific activity was blocked by 2 h incubation at room temperature with 10% bovine serum albumin in a solution of 0.2% Triton X-100 in PBS. Solubilization of cell membranes was carried out during 1 h incubation at room temperature in a solution of 1% Triton X-100 in PBS. Incubation with primary antibodies in a 1:500 dilution took place overnight at 4 °C with rabbit anti-Aβ 1–42 peptide antibody (1:200; no. MAA946Ge21, Cloud Clone, Wuhan, China) and mouse anti- NG2 antibody (1:500; ab273464, Abcam, Waltham, MA, USA). At all stages, the samples were washed 3–4 times for 5 min incubation in a washing solution. After that, the corresponding secondary antibodies goat anti-rabbit IgG (H+L) Alexa Flour 488, goat anti-rabbit IgG (H+L) Alexa Flour 555, and goat anti-mouse IgG (H+L) Alexa Flour 555 (Invitrogen, Molecular Samples, Eugene, OR, USA) were applied. At the final stage, the sections were transferred to the glass and 15 µL of mounting liquid (50% glycerin in PBS) was applied to the section. The slices were covered with a cover glass and confocal microscopy was performed.

The sections of brains, dcLNs, and meninges were visualized using a confocal microscope LEICA TCS SP8 (Leica Microsystems, Wetzlar, Germany) with a ×20 lens (0.75 NA) or a ×100 lens for immersion in oil (0.45 NA). DAPI, Alexa Fluor 488, and Alexa Fluor 555 were excited with excitation wavelengths of 405 nm, 488 nm, and 561 nm, respectively. Evans Blue were excited with the same excitation wavelength of 647 nm. Three-dimensional visualization data were collected by software 3.0.16120.2 LAS X (Leica Microsystems, Germany) and analyzed using Fiji software 2.0.0 (Open-source image processing software).

### 4.6. ELISA Analysis of Brain Tissues

For ELISA, a kit for the determination of the soluble form of Aβ (1–42) (Cloud Clone, no. CEA946Mu) was used. In mice from all the studied groups, the brains were collected, and further sample preparation was carried out. For subsequent ELISA, brain tissues were homogenized and lysed for further peptide isolation and preparations using Cloud Clone Protocol. In total, lysates from brain tissues were prepared in a lysing buffer (1.5 mm KH_2_PO_4_, 8 mm Na_2_HPO_4_, 3 mm KCl, 137 mm NaCl, and 0.1% Twin20, 10 mM EDTA), pH 7.2, with a freshly prepared protease inhibitory mixture (Roche Applied Science, Penzberg, Germany). Measurements of the optical density of the studied samples were carried out at a wavelength of 450 nm (A450) on an automatic enzyme immunoassay analyzer–microplate spectrophotometer Epoch BioTek Instrument (BioTek Instruments, Vantaa, Finland). The obtained results were subjected to statistical processing. Confidence intervals were determined for 95% of the significance level. The processing of experimental data with a sample size (n = 8–10) was carried out by the method of univariate analysis of variance ANOVA.

### 4.7. Statistical Analysis

All statistical analyses performed using ImageJ (Version 1.54k), Microsoft Office Excel, and STATISTICA 10 for Windows software. The results were reported as a mean value ± standard error of the mean (SEM). The differences in the signal intense of Aβ in the tested tissues were evaluated using the ANOVA test with post hoc Duncan test and Mann–Whitney U test. The significance levels were set at *p* < 0.05 for all analyses. No statistical methods were used to predetermine sample size.

## 5. Conclusions

In summary, our findings provide compelling evidence that Pls could be a promising candidate to enhance the lymphatic clearance of the soluble forms of Aβ from the aging brain and in AD subjects to improve cognitive functions. Indeed, a 21-day course of Pls administration into the right lateral ventricle significantly reduces the elevated Aβ brain levels in aged (14-month-old) mice and in animals with the early stage of AD. Pls effectively facilitate learning, i.e., developing a conditioned reflex, in AD mice as well as in young, middle-aged, and aged animals. However, Pls show no effects on both Aβ clearance and learning in old (24-month-old) mice. Thus, Pls can be used as an adjuvant and alternative therapy for age-related changes in the brain (taking into account the limited effects of Pls at the late stages of ontogenesis) and early forms of AD associated with the excessive accumulation of soluble forms of Aβ in CNS.

## Figures and Tables

**Figure 1 ijms-25-12552-f001:**
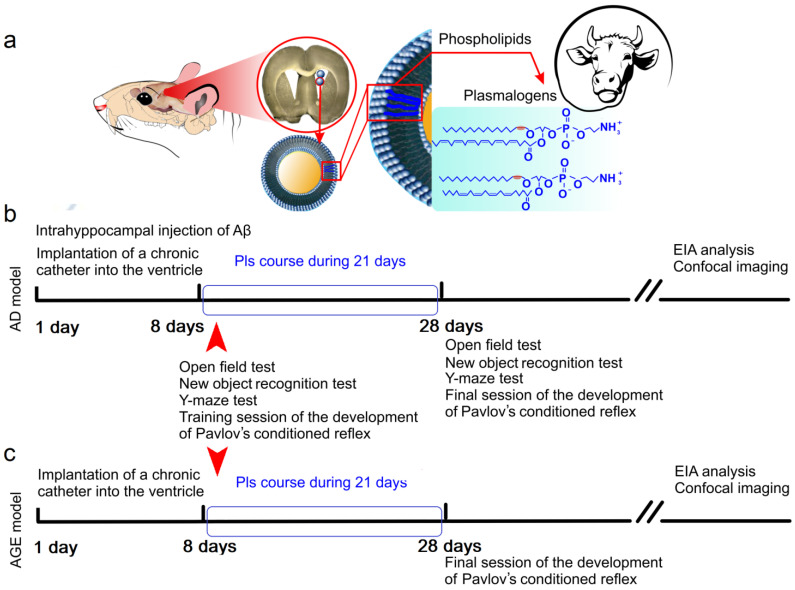
Schematic illustration of the study design: (**a**) Pls (phospholipids from bovine brain) were administered for 21 days into the right lateral ventricle through a chronic catheter; (**b**,**c**) the effects of Pls on clearance of the soluble forms of Aβ and cognitive functions were studied in two functional models, including (**b**) an injection model of early AD (unilateral administration of Aβ into the hippocampus) and (**c**) the age model reflecting natural age-related changes in Aβ deposition in brain tissues. Before and after a 21-day course of Pls, studies of behavior of mice as well as qualitative and quantitative analysis of Aβ in brain tissues were conducted using enzyme immunoassay (EIA) and confocal imaging of Aβ in the brain, the meninges, and in the deep cervical lymph nodes (dcLNs).

**Figure 2 ijms-25-12552-f002:**
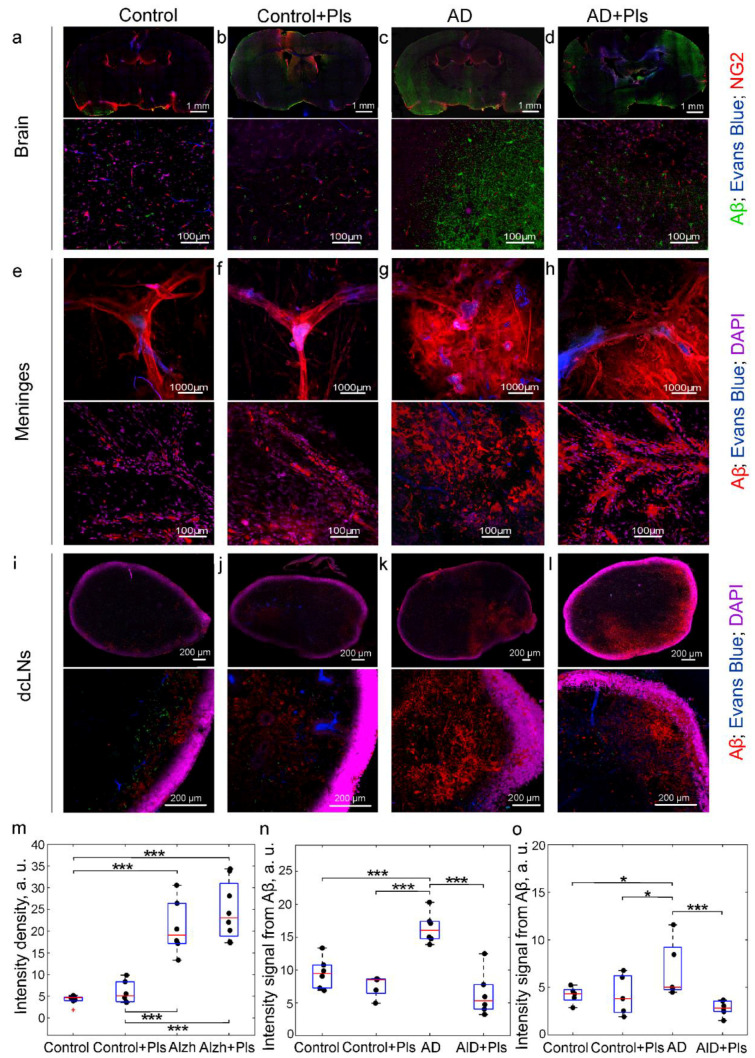
The effects of Pls on Aβ clearance from the brain and the meninges to the peripheral lymphatics in AD mice: (**a**–**d**) Representative images of Aβ (green) in the brain from the tested groups. The blood vessels are filled with Evans Blue (blue) and labeled with NG2 (red); (**e**–**h**) Representative images of Aβ (red) in the meninges from the tested groups; (**i**–**l**) Representative images of Aβ (red) in dcLNs from the tested groups. In (**e**–**l**), the blood vessels are filled with Evans Blue (blue), the cell nuclei are labeled with DAPI (violet); (**m**–**o**) Quantitative analysis of the intensity of fluorescent signal from Aβ labeled with primary and secondary antibodies in the brain (**m**), the meninges (**n**), and in dcLNs (**o**), n = 7 in each group, *—*p* < 0.05, ***—*p* < 0.001, the ANOVA test with the post hoc Duncan test.

**Figure 3 ijms-25-12552-f003:**
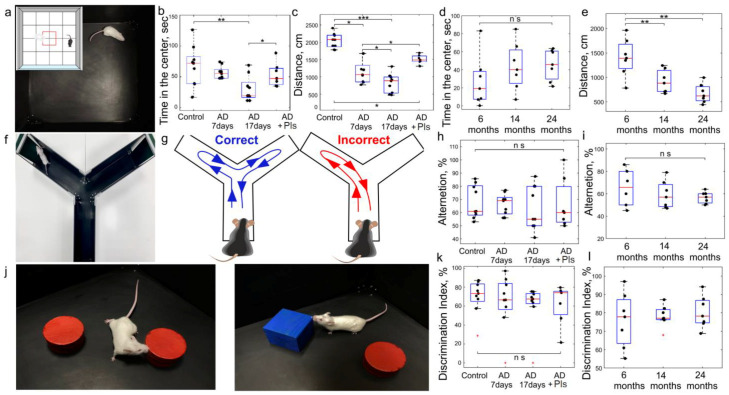
The effects of Pls on cognitive functions in AD mice and mice of different ages: (**a**–**e**) Assessment of locomotor activity and anxiety using the open-field test (**a**) in the AD groups (**b**,**c**) without Pls and after a 21-day course of Pls as well as in middle-aged, aged, and old mice without Pls (**d**,**e**); (**f**–**i**) Evaluation of spatial memory using the Y-maze test (**f**,**g**) in the AD groups (**h**) without Pls and after a 21-day course of Pls and well as in middle-aged, aged, and old mice without Pls (**i**); Analysis of recognition memory using the new object recognition test (**j**) in the AD groups (**k**) without Pls and after a 21-day course of Pls as well as in middle-aged, aged, and old mice without Pls (**l**); n = 8 in the AD groups and n = 7 in the age groups, *—*p* < 0.05, **—*p* < 0.01, ***—*p* < 0.001, ns means not significant, the ANOVA with the post hoc Tukey HSD Test.

**Figure 4 ijms-25-12552-f004:**
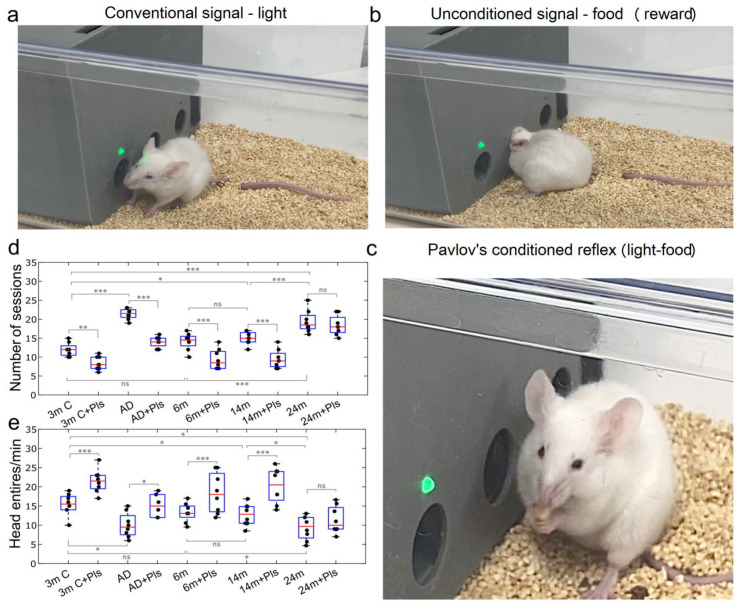
The effects of Pls on the development of Pavlov’s conditioned reflex in AD mice and mice of different ages: (**a**–**c**) the training of mice is based on the presentation of an unconditional signal (green light), after which the animal receives food (reward) if it accidentally finds the correct window, in which food (seeds) falls out when the head is howled. The number of training sessions required for the mouse to form a stable conditioned reflex is assessed (when the light is turned on, the mouse must quickly (within 15 s) find the correct window and receive food (reward)). The number of rewards (food) received is also assessed in the final session; (**d**,**e**) Assessment of Pavlov’s conditioned reflex in AD mice and mice of different ages, n = 8 in each group, *—*p* < 0.05, **—*p* < 0.01, ***—*p* < 0.001, ns means not significant, the ANOVA test with the post hoc Duncan test.

**Figure 5 ijms-25-12552-f005:**
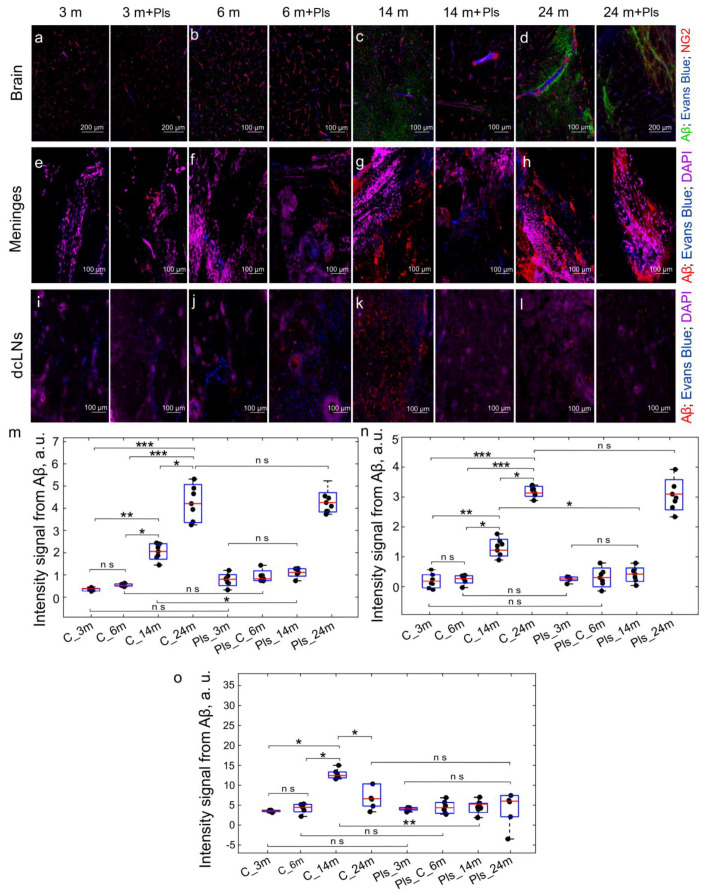
The effects of Pls on Aβ clearance from the brain and the meninges to the peripheral lymphatics in AD mice: (**a**–**d**) Representative images of Aβ (green) in the brain from the tested groups. The blood vessels are filled with Evans Blue (blue) and labeled with NG2 (red); (**e**–**h**) Representative images of Aβ (red) in the meninges from the tested groups; (**i**–**l**) Representative images of Aβ (red) in dcLNs from the tested groups. In (**e**–**l**), the blood vessels are filled with Evans Blue (blue), the nuclei are labeled with DAPI (violet); (**m**–**o**) Quantitative analysis of Aβ levels in the brain (**m**), the meninges (**n**), and in dcLNs (**o**), n = 7 in each group, *—*p* < 0.05, **—*p* < 0.01, ***—*p* < 0.001, ns means not significant, the Mann–Whitney U Test.

**Table 1 ijms-25-12552-t001:** The level of soluble Aβ (pg/g tissue) in the brain of the tested groups without and after the Pls therapy.

Tested Groups	no Pls	+Pls
AD mice	23.43 ± 1.14 ^###^	16.07 ± 2.14 ^†^
3-month-old mice	10.18 ± 1.23	10.04 ± 1.32
6-month-old mice	11.07 ± 1.01	11.12 ± 1.03
14-month-old mice	14.29 ± 1.05 **	11.00 ± 1.01 ^†^
24-month-old mice	18.83 ± 1.07 ***	17.95 ± 1.86

**—*p* < 0.001; ***—*p* < 0.001 between age groups; ^†^—*p* < 0.05 between the groups received and not Pls; ^###^—*p* < 0.001 between AD and healthy 3-month-old mice; n = 7 in each group; the ANOVA test with post hoc Duncan test.

## Data Availability

The original contributions presented in this study are included in the article; further inquiries can be directed to the corresponding authors.

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
