# Peer review of "Plasmalogens Improve Lymphatic Clearance of Amyloid Beta from Mouse Brain and Cognitive Functions"

_ijms, 2024, doi:10.3390/ijms252312552_

Round 1
Reviewer 1 Report
Comments and Suggestions for Authors
I found this a very interesting paper and was learned something new in the potential importance about plasmologens which I have to admit not knowing much about yet I was really interested to read this submission.
I have some minor comments and also a few slightly more involved suggestions.
Minor:
- It may sound pedantic but the amyloid is usually described as a peptide rather than a protein because the traditional demarcation between the two is usually smaller proteins, i.e. <50 amino acids are usually referred to as peptides (line 45 and check elsewhere for consistency).
- line 113 in the legend - to improve clarity add 'b' before "..a model of the age model...."
- line 504- in the comments about AD patients and car keys - I am not entirely sure I agree with the statement that an AD patient would not remember the purpose of car keys, maybe more advanced patients yes, however I suspect many might be more likely to remember the keys and what they are for but might have failing recollection in the identification of their car. My suggestion is to remove this as an example of early signs of memory loss - I do not think it is appropriate here and in fact might be likely to cause a lot of offence to many people.
Major:
Lines 73-74 - the description to Aducanumab, while accurate is now somewhat out of date, since the company Biogen are no longer pursuing it through approvals while new drugs Lecanumab and Donanemab which have since been reported are now going through approvals - this section of the introduction therefore needs updating.
Figure 1 - I was a little confused about whether Figure 1 was intended as a graphical abstract of sorts or as a Figure - while i appreciate the nice artistic slant in some of the images provided I felt some of them (e.g. the images to denote AD model and aged model) did not really add anything and if anything were a distraction as their purpose was somewhat unclear. For me the information provided in parts c, g and h are the most valuable and important to the paper and suggest simplifying the image somewhat - the images a, b, d, e, and f don't really add value I am sorry to say.
line 138 - I was a little surprised to see that the plasmologens made here were derived from bovine sources without some justification as to why? Is it simply due to availability or ease of access or due to close similarities to human forms? On a related note, considering the particularly difficult history of bovine brain disease and later in humans - in a study seeking to extol the virtues of a new treatment, that these might be derived from a bovine source, the cause of human variant CJD in humans would not go well from a marketing perspective and some comment on this would be useful in the discussion regarding alternative sources that might also be amenable to being used?
lines 323-326 - it looks like from the descriptions given of the preparation of tissue for the ELISAs that the abeta species being measured is likely to be soluble abeta species? This makes sense given that clearance processes are being assessed yet it might have been a missed opportunities not to have also had an insoluble Abeta homogenisation protocol and this could have provided another amount of data to support whether clearance was being suppported. I imagine going back to do additional experiments may be a challenge and so might having any tissue remaining to be able to do such work so at the very least I think the authors could clarify what Abeta species their measurements were detecting and then having commenting on the possible option in future studies to include and insoluble abeta measurement option as well?
- Conclusions section: I think this conclusions paragraph, while it contains a lot of valid information is too long and I believe there is about half of its content that would be better served in a section at the end of the discussion, before the Conclusions that is entitled Strengths and limitations. Some of the questions posed in the conclusion would be better placed here. To this section it could also contain some brief discussion about the selection of 'bovine' as a source for the plasmologens, and the abeta species measured and one other limitation, which I think was a missed opportunity was to have done an AD model at the 13-14 month age mark as well because modelling 'early AD' in 3 month old mice is actually more like modelling early AD in young people whereas the reality is that older mice with the AD modelled might have been closer to testing the potential therapeutic benefit of Pls and I think this should be stated as a clear limitation and a suggestion for future work. Eventhough the authors models that age group in the aged cohort - what cannot be excluded is whether there any other age-related changes that might affect the sensitivity of the mice to the abeta exposure a year later than was tested and so a closer approximation of Pls as a treatment could have been modelled.
Comments on the Quality of English LanguageThere are a small number of places where small adjustments to the grammar need to be done
see lines:
89 - large amount should be plural
206 -intrahyppocampal should be intrahippocampal
215 - is byclean = by clean?
437 - 'was not differ' should be either 'did not differ' or 'were not different' (because describing 3 areas i.e. plural)
Author Response
I found this a very interesting paper and was learned something new in the potential importance about plasmologens which I have to admit not knowing much about yet I was really interested to read this submission. I have some minor comments and also a few slightly more involved suggestions.
Comment: It may sound pedantic but the amyloid is usually described as a peptide rather than a protein because the traditional demarcation between the two is usually smaller proteins, i.e. <50 amino acids are usually referred to as peptides (line 45 and check elsewhere for consistency).
Response: The authors would like to express their sincere gratitude to the reviewer for constructive advice and recommendations to improve the quality of our article. We agree that beta-amyloid is a peptide and have therefore made corrections to the text (line 45). All changes in the article are highlighted in yellow.
Comment: line 113 in the legend - to improve clarity add 'b' before "..a model of the age model...."
Response: We have corrected Figure 1 and its legend accordingly (Lines 144-152).
Comment: line 504- in the comments about AD patients and car keys - I am not entirely sure I agree with the statement that an AD patient would not remember the purpose of car keys, maybe more advanced patients yes, however I suspect many might be more likely to remember the keys and what they are for but might have failing recollection in the identification of their car. My suggestion is to remove this as an example of early signs of memory loss - I do not think it is appropriate here and in fact might be likely to cause a lot of offence to many people.
Response: We have removed this sentence from the text.
Comment: Lines 73-74 - the description to Aducanumab, while accurate is now somewhat out of date, since the company Biogen are no longer pursuing it through approvals while new drugs Lecanumab and Donanemab which have since been reported are now going through approvals - this section of the introduction therefore needs updating.
Response: We have added this information to the introduction (Lines 103-105).
Comment: Figure 1 - I was a little confused about whether Figure 1 was intended as a graphical abstract of sorts or as a Figure - while i appreciate the nice artistic slant in some of the images provided I felt some of them (e.g. the images to denote AD model and aged model) did not really add anything and if anything were a distraction as their purpose was somewhat unclear. For me the information provided in parts c, g and h are the most valuable and important to the paper and suggest simplifying the image somewhat - the images a, b, d, e, and f don't really add value I am sorry to say.
Response: We have corrected Figure 1.
Comment: line 138 - I was a little surprised to see that the plasmologens made here were derived from bovine sources without some justification as to why? Is it simply due to availability or ease of access or due to close similarities to human forms? On a related note, considering the particularly difficult history of bovine brain disease and later in humans - in a study seeking to extol the virtues of a new treatment, that these might be derived from a bovine source, the cause of human variant CJD in humans would not go well from a marketing perspective and some comment on this would be useful in the discussion regarding alternative sources that might also be amenable to being used?
Response: Yes, we used this source due to close similarities to human forms. The fatty acid composition of marine Pls is very different from that of mammals. And it's one thing when Pls enter the stomach and are metabolized there, and quite another when they are injected directly into the ventricle of the brain. Therefore, in our opinion, leaving the marine source in this case is quite adequate.
Comment: lines 323-326 - it looks like from the descriptions given of the preparation of tissue for the ELISAs that the abeta species being measured is likely to be soluble abeta species? This makes sense given that clearance processes are being assessed yet it might have been a missed opportunities not to have also had an insoluble Abeta homogenisation protocol and this could have provided another amount of data to support whether clearance was being suppported. I imagine going back to do additional experiments may be a challenge and so might having any tissue remaining to be able to do such work so at the very least I think the authors could clarify what Abeta species their measurements were detecting and then having commenting on the possible option in future studies to include and insoluble abeta measurement option as well?
Response: We have revised the methods and added to the limitations of our studies in determining the Pls effects on soluble forms of Aβ and the prospects for further study of the effects of Pls on insoluble forms of Aβ (Lines 146, 333, 367, 448, 476, 621, 652, 609-638).
Comment: Conclusions section: I think this conclusions paragraph, while it contains a lot of valid information is too long and I believe there is about half of its content that would be better served in a section at the end of the discussion, before the Conclusions that is entitled Strengths and limitations. Some of the questions posed in the conclusion would be better placed here. To this section it could also contain some brief discussion about the selection of 'bovine' as a source for the plasmologens, and the abeta species measured and one other limitation, which I think was a missed opportunity was to have done an AD model at the 13-14 month age mark as well because modelling 'early AD' in 3 month old mice is actually more like modelling early AD in young people whereas the reality is that older mice with the AD modelled might have been closer to testing the potential therapeutic benefit of Pls and I think this should be stated as a clear limitation and a suggestion for future work. Eventhough the authors models that age group in the aged cohort - what cannot be excluded is whether there any other age-related changes that might affect the sensitivity of the mice to the abeta exposure a year later than was tested and so a closer approximation of Pls as a treatment could have been modelled.
Response: We have added the study limitations and corrected the conclusion (Lines 609-638, 651-661).
Comments: There are a small number of places where small adjustments to the grammar need to be done, see lines:
89 - large amount should be plural
206 -intrahyppocampal should be intrahippocampal
215 - is byclean = by clean?
437 - 'was not differ' should be either 'did not differ' or 'were not different' (because describing 3 areas i.e. plural)
Response: We have corrected English and grammar errors throughout the article.
The authors thank the reviewer once again for the great help to improve of the quality of the article with important comments for possible publication in the International Journal of Molecular Sciences.
Authors

Reviewer 2 Report
Comments and Suggestions for Authors
Shirokov et al. manuscript provides compelling insights on plasmalogens (PLs) as potential candidates for AD and age-related brain disease due to their positive impact on amyloid beta (Aβ) clearance. Considering the increasing burden – from both a personal and economic point of view – represented by neurodegenerative and age-related diseases, the discovery and the development of potential novel therapeutic approaches is of paramount interest. Therefore, the topic of the manuscript is timely and the reported findings could be of great interest to the field and the readership. The abstract provides a clear and focused recap of the authors’ work. The materials and methods section is clear and well presented. However, despite my positive opinion on the present article, I am of the opinion that some minor and major concerns hinder its publication in the present form. Hence, to ensure publication suitability, this reviewer raises these major and minor issues as detailed below:
Major issues:
1. Despite being generally well-structured, the introduction part lacks pivotal information to guarantee a correct presentation of the state of the art necessary to comprehend the manuscript. A brief explanation of APP processing resulting in Aβ formation (or not) should be added (e.g. 10.3390/ijms24076639) for a complete overview on APP/Aβ background.
2. The presentation of the neurophysiologic roles of Aβ (lines 50-52) should be deepened (e.g. 10.2174/13816128113199990503; 10.3389/fnagi.2014.00166; 10.3233/JAD-2010-1299; 10.1038/sj.npp.1301485; 10.1371/journal.pone.0029661; 10.3389/fphar.2012.00146; 10.1016/j.ejphar.2013.11.011), since in the present state appears quite dismissive.
3. In lines 92-94, the authors state that “Aβ reduces PLs levels in the brain, while the decreased PLs levels directly increase γ-secretase activity leading to stronger production of Aβ”. Considering that γ-secretase operates downstream either α-secretase and BACE1 (and limiting only to the two major APP processing pathways), this statement is not correctly supported by literature. Knowing that Aβ production could be the result of either BACE1 or other APP amyloidogenic processing, it should be expected that PLs deficiency may upregulate/increase the activity of BACE1 (or other Aβ-producing secretases) rather than γ-secretase. Can the authors explain their line of thought on this matter in a more detailed way? Are PLs reported to modulate BACE1 expression/activity? Moreover, a mechanistic explanation of how PLs can improve AD-related molecular aspects is missing and should be implemented for a comprehensive presentation of the state of the art.
4. In lines 101-103, the authors reported that PLs oral administration improved cognitive function and mental condition of AD patients. Can the authors explain why they chose catheter-assisted right lateral ventricle injection as PLs mode of administration? Why oral PLs administration was not considered?
5. The authors declared that male C57BL/6 mice were employed in all experiments. Considering the increasing knowledge on sex and gender differences in AD, particularly for women, also correlated with age (e.g. 10.1016/j.mad.2023.111821; 10.14283/jpad.2018.34; 10.1016/j.jalz.2018.04.008), are the authors aware of any sex-related differential effects of PLs administration? Could the choice of excluding female mice have introduced some sort of bias in the experimental outcome? If so, this should be properly addressed in a “limitations of the study” part that must be added.
6. Considering that the deposition/formation of Aβ fibrils could be the result of different underlying impairments that Aβ injection alone may not mimic, the AD model here employed could appear biased (despite being correctly referenced). Are the authors aware (or have the possibility to perform experiments on) of PLs in other well-established AD models (e.g. 3xTg)?
7. Was the ELISA analysis of brain tissues (paragraph 2.6) performed on total brains or specific brain areas? Considering the importance of hippocampus, prefrontal cortex and entorhinal cortex in AD development and progression, can these (and the other) analyses be performed on these specific brain areas?
8. The authors point at age-related MLVs dysfunctions (lines 450; 464-465) as a possible contributing factor for the reduced PLs effects on Aβ clearance. Can the authors substantiate this statement (e.g. showing increased/decreased markers of MLVs dysfunction that correlate with AD and/or age and to explain why the effect was visible only in 14 m.o. mice)? Otherwise this aspect remains speculative.
Minor issues:
1. Consistency in how things have been abbreviated is lacking (e.g. Aβ(1-42) vs Aβ42; PLs vs Pls; etc). Please ensure that abbreviations are consistent throughout the text.
2. Immunohistochemical assay should be properly referenced.
3. Panel order in the figure must reflect their order of presentation in the text.
4. Graphs/tables reporting ELISA results must be included rather than made explicit in the text for a correct data presentation.
5. What do the authors mean with “knockdown of PLs” (line 540)? How is it possible to knockdown a phospholipid?
Comments on the Quality of English LanguageThe authors should submit their manuscript for a deep proof-reading from a native speaker to ensure the correct use of the English language. Several sentences throughout the text do not sound natural and need rearranging. In addition, punctuation and typos (e.g. “synaptic vehicles” instead of “synaptic vesicles”, “special memory” instead of “spatial memory”, “-catenin” instead of “β-catenin”) must be checked and other minor editing should be performed on the entire manuscript.
Author Response
Shirokov et al. manuscript provides compelling insights on plasmalogens (PLs) as potential candidates for AD and age-related brain disease due to their positive impact on amyloid beta (Aβ) clearance. Considering the increasing burden – from both a personal and economic point of view – represented by neurodegenerative and age-related diseases, the discovery and the development of potential novel therapeutic approaches is of paramount interest. Therefore, the topic of the manuscript is timely and the reported findings could be of great interest to the field and the readership. The abstract provides a clear and focused recap of the authors’ work. The materials and methods section is clear and well presented. However, despite my positive opinion on the present article, I am of the opinion that some minor and major concerns hinder its publication in the present form. Hence, to ensure publication suitability, this reviewer raises these major and minor issues as detailed below:
Comments: Despite being generally well-structured, the introduction part lacks pivotal information to guarantee a correct presentation of the state of the art necessary to comprehend the manuscript. A brief explanation of APP processing resulting in Aβ formation (or not) should be added (e.g. 10.3390/ijms24076639) for a complete overview on APP/Aβ background.
The presentation of the neurophysiologic roles of Aβ (lines 50-52) should be deepened (e.g. 10.2174/13816128113199990503; 10.3389/fnagi.2014.00166; 10.3233/JAD-2010-1299; 10.1038/sj.npp.1301485; 10.1371/journal.pone.0029661; 10.3389/fphar.2012.00146; 10.1016/j.ejphar.2013.11.011), since in the present state appears quite dismissive.
Response: The authors would like to thank the reviewer for the important advices and recommendations to improve the quality of our article. We have proofread the introduction according to the comments (Lines 44-73, 77-81, 126-141). All changes are highlighted in yellow.
Comment: In lines 92-94, the authors state that “Aβ reduces PLs levels in the brain, while the decreased PLs levels directly increase γ-secretase activity leading to stronger production of Aβ”. Considering that γ-secretase operates downstream either α-secretase and BACE1 (and limiting only to the two major APP processing pathways), this statement is not correctly supported by literature. Knowing that Aβ production could be the result of either BACE1 or other APP amyloidogenic processing, it should be expected that PLs deficiency may upregulate/increase the activity of BACE1 (or other Aβ-producing secretases) rather than γ-secretase. Can the authors explain their line of thought on this matter in a more detailed way? Are PLs reported to modulate BACE1 expression/activity? Moreover, a mechanistic explanation of how PLs can improve AD-related molecular aspects is missing and should be implemented for a comprehensive presentation of the state of the art.
Response: The aim of our studies was to investigate the possibility of stimulation of lymphatic removal of Aβ from brain tissues of mice with AD and of different ages using Pls. The choice of this goal is related to the fact that meningeal lymphatic vessels (MLVs) are “tunnels” for removal of Aβ from the brain. Since we expected that Pls could reduce the level of soluble Aβ in the brain, we assumed that this could be due to Pls-mediated increase in the MLV functions. We have corrected the text of the article to make more clearly the main idea of our studies. We also added in the limitations a discussion of the need to study the effects of Pls on the main mechanisms of Aβ production, including the APP and BACE1 pathways.
Comment: In lines 101-103, the authors reported that PLs oral administration improved cognitive function and mental condition of AD patients. Can the authors explain why they chose catheter-assisted right lateral ventricle injection as PLs mode of administration? Why oral PLs administration was not considered?
Response: The choice of intraventricular administration of Pls is explained by the goal of studying the effects of Pls on the removal of soluble forms of amyloids, i.e. dissolved in the cerebrospinal fluid (CSF). Despite the growing number of publications indicating the possibility of using Pls to improve cognitive functions in patients and animals with Alzheimer's disease, these studies are of a phenomenological nature. It remains unknown whether Pls can penetrate the blood-brain barrier (BBB) or have an indirect therapeutic effect. It is not clear how Pls are transported from the gastrointestinal tract to the brain and what is their bioavailability when administered orally. The introduction of Pls directly into the cerebrospinal fluid system of the brain allows us to directly answer the question of whether Pls has an effect on the removal of the soluble forms of amyloid from CSF. In the future, we plan to study similar effects with oral administration of Pls, as well as to study in detail their passage through BBB. We have added this explanation in the limitations of studies (Lines 609-649).
Comment: The authors declared that male C57BL/6 mice were employed in all experiments. Considering the increasing knowledge on sex and gender differences in AD, particularly for women, also correlated with age (e.g. 10.1016/j.mad.2023.111821; 10.14283/jpad.2018.34; 10.1016/j.jalz.2018.04.008), are the authors aware of any sex-related differential effects of PLs administration? Could the choice of excluding female mice have introduced some sort of bias in the experimental outcome? If so, this should be properly addressed in a “limitations of the study” part that must be added.
Response: We added the absence of these data in our article to the limitations (Lines 609-649). The absence of results on sex differences in our studies is due to the aim of focusing experiments on the investigations of the role of MLVs in the stimulating effects of Pls on the lymphatic removal of soluble Aβ from the brain. No gender differences were found in the MLVs functions in mature female and male mice (Nature 2019, 572, 62-66). Therefore, our research was not focused on studying sex differences in the therapeutic effects of Pls. We assume that Pls improves lymphatic removal of Aβ from the brain equally in both reproductive male and female. Nevertheless, our data on the decrease in sensitivity to Pls therapy with age make it relevant to study sex differences in the effectiveness of Pls in the elderly.
Comment: Considering that the deposition/formation of Aβ fibrils could be the result of different underlying impairments that Aβ injection alone may not mimic, the AD model here employed could appear biased (despite being correctly referenced). Are the authors aware (or have the possibility to perform experiments on) of PLs in other well-established AD models (e.g. 3xTg)?
Response: The choice of the injected model of AD is related to the fact that this model allows to study the impairment of cognitive functions associated with the direct introduction of Aβ into the hippocampus. Therefore, the different models of injection of Aβ into the hippocampus or in the brain ventricles are often used in the studies of early forms of AD (Acta Neuropathol Commun 2022, 10, 113; Cytotherapy 2019, 21(6):671-682; Methods Mol Biol. 2018, 1727, 343-352; Cells 2023, 12, 694). Since our studies were focused on analysis of the effects of Pls on lymphatic removal of soluble forms of Aβ from the brains, the injection model of AD is most suitable for this purpose. However, the study of the therapeutic effects of Pls on other AD models, including 5xFAD, APP/PS1, 3xTg etc., is a necessary step in assessment of reproducibility of the results. We added this consideration to the limitations (Lines 609-649).
Comment: Was the ELISA analysis of brain tissues (paragraph 2.6) performed on total brains or specific brain areas? Considering the importance of hippocampus, prefrontal cortex and entorhinal cortex in AD development and progression, can these (and the other) analyses be performed on these specific brain areas?
Response: Measurement of soluble Aβ levels was performed in the whole brain. The choice of these tested tissues was related to the purpose of studying the effects of Pls on the lymphatic removal of soluble Aβ from the central nervous system. For this purpose, the choice of individual brain regions for determining the Aβ levels was inappropriate because we were answering the question of whether Pls can promote the lymphatic removal of soluble Aβ, i.e. the removal of Aβ dissolved in the cerebrospinal fluid (CSF), which drains Aβ from brain tissues into its meninges and then into dcLNs - the first anatomical station for collection of CSF with substances dissolved. We have added this information to the methods to make the choice of methods and objects for our studies clearer (Lines 330-337).
Comment: The authors point at age-related MLVs dysfunctions (lines 450; 464-465) as a possible contributing factor for the reduced PLs effects on Aβ clearance. Can the authors substantiate this statement (e.g. showing increased/decreased markers of MLVs dysfunction that correlate with AD and/or age and to explain why the effect was visible only in 14 m.o. mice)? Otherwise this aspect remains speculative.
Response: Age-related decline in the MLV functions has been studied in detail and published in these papers (Nature 2018, 560(7717), 185-191; Nature 2019, 572(7767), 62-66). It is a known fact that the morphology of MLVs changes with age. The process of lymphatic hyperplasia and lymphatic valve dysfunction begins to manifest in aging mice (starting from 13-14 months of age) and progresses to old age leading to a dramatically reducing brain drainage. In our other paper in the Frontiers of Optoelectronics, which has been accepted for publication but has not yet been published, we also found lymphatic hyperplasia in 24-month-old mice, which is associated with low brain’s drainage as well as with the lack of sensitivity to stimulating photo-effects on lymphatic removal of Aβ from the brain. The lack of of Pls effects on the lymphatic removal of Aβ may be associated with age-related changes in the MLV functions. We have clearly shown in 2 functional models - the early form of AD and the aging brain that Pls do not have effects on the removal of Aβ from the brain to the meninges and the deep cervical lymph nodes. We agree that age-related morphological changes in MLVs are not directly demonstrated in our studies. However, these data are well presented in other articles and therefore we refer to already known facts and put forward a hypothesis that age-related changes in the structure and functions of MLVs may be one of the reasons explaining the absence of stimulating effects of Pls on lymphatic removal of Aβ from the brain in old mice (Lines 582-608).
Comments:
Consistency in how things have been abbreviated is lacking (e.g. Aβ(1-42) vs Aβ42; PLs vs Pls; etc). Please ensure that abbreviations are consistent throughout the text.
Immunohistochemical assay should be properly referenced.
Panel order in the figure must reflect their order of presentation in the text.
Graphs/tables reporting ELISA results must be included rather than made explicit in the text for a correct data presentation.
What do the authors mean with “knockdown of PLs” (line 540)? How is it possible to knockdown a phospholipid?
The authors should submit their manuscript for a deep proof-reading from a native speaker to ensure the correct use of the English language. Several sentences throughout the text do not sound natural and need rearranging. In addition, punctuation and typos (e.g. “synaptic vehicles” instead of “synaptic vesicles”, “special memory” instead of “spatial memory”, “-catenin” instead of “β-catenin”) must be checked and other minor editing should be performed on the entire manuscript.
Response: We have corrected all grammatical and stylistic errors, as well as the English language throughout the text of the article.
The authors would like to express their sincere gratitude for the in-depth review of our article and important recommendations for its possible publication in the International Journal of Molecular Sciences.
Authors

Reviewer 3 Report
Comments and Suggestions for Authors
I have read with interest the manuscript “Plasmalogens improve clearance of amyloid-beta from mouse brain and cognitive functions” by Shikorov et al. The authors address the potentially beneficial therapeutic effect of plasmalogens against the deposition of amyloid beta. They employ non-transgenic mice with two main series of experiments, either by injecting recombinant amyloid-beta fibrils or by following the endogenous amyloid-beta accumulation. In both cases they treat the mice with plasmalogens and assess the effect by ELISA, immunofluorescence, and behavioural tests.
Overall, the study is interesting and obviously the use of in vivo models is of higher importance and amount of work. However, there are several problems throughout the manuscript that need to be addressed before proceeding for publication.
1. First of all, there are many grammatical errors especially in the introduction, but also in other parts of the text. I listed some of them at the end of my comments and I encourage the authors to correct them and carefully proof-read the manuscript.
2. The mice employed for this study are non-transgenic mice. However, the authors demonstrate a time-dependent accumulation of Aβ42 (Figure 5). Is this a finding that has been previously reported by other groups? I tried to find citations in the manuscript supporting that, but I was not able to locate them. I find this quite confusing, because there are several transgenic mouse models of Aβ deposition in the literature, so I am not sure whether this is a known feature for mice to deposit Aβ42 naturally over time. If they do, what is the homology between the human and murine version? Have the authors addressed biochemically or by stainings the nature of these Aβ42 species? Perhaps detergent-insoluble centrifugation assays or stainings with amyloid dyes could help.
3. The authors report their systems as AD or injected AD based on injections and IF stainings of Aβ42. I would argue that this is not the case, because to classify human patients as AD cases there needs to be the concurrent deposition of Aβ plaques and Tau neurofibrillary tangles. Additionally, the filamentous structures of these amyloids are very specific and reported in detail over the last years (PMID: 28678775, 35025654). Therefore, the models mentioned here only partially replicate the AD situation and this is something that should be mentioned. Perhaps the authors should mention them as AD-like mouse models, since they do not recapitulate the main aspects of AD pathology.
4. There are also several issues with the presentation of the findings.
a. Even though I appreciate the effort of the authors to visualise the design of their experimental approach, the schematic of Figure 1 is overloaded with lots of information and needs to be simplified. The artistic drawings in Figure 1a and b should also be removed, as they are not related to the nature of the experiments. Additionally, what does EIA mean and why the authors do not use the term ELISA? Overall, I think that the schematics of Figure 1g and h are enough to illustrate the experiment.
b. Figure 2: Multiple issues that must be addressed:
i. Why did the authors perform NG2 labelling for the brain samples instead of DAPI for the rest of the regions? (Note: the same question applies to Figure 5). The authors also do not report the antibody for the NG2 staining.
ii. The presentation is rather poor. Starting with the brain images, the control top left image seems to have very high levels of Aβ, way higher than the injected samples. Additionally, for the zoomed areas the authors should present images from areas that are similar in all cases. While the control zoomed images are similar, the AD and the AD + Pls are from completely different areas. The authors should improve this, because it is important to display representative images that are in line with the quantified data.
iii. The scale bars are all in different places in the panels, there needs to be concistency on that point so that it is more presentable. Additionally, why are the scale bars of the AD samples for the dcLNs different compared to the others?
c. Figure 5: The dcLNs representative images are very different from the equivalent ones in Figure 2. Why is that and how do the authors comment on it?
5. Lines 359-364, 406-413: very confusing comparison for the ELISA results as they describe them in the text, almost impossible to follow. The authors should either plot them in graph or in table. Additionally, the levels of Aβ in the uninjected animals are estimated at about 10 pg/g tissue and after injection it goes up to 23.4 pg/g tissue, while in the 24-month-old uninjected mice the levels are about 18 pg/g tissue. To me this sounds weird, as I would expect the injected animals to have way higher levels of Aβ.
Specific comments for grammar errors:
Line 45: “which is involved in pathogenesis”, missing the word “the”
Lines 48: “result of the cleavage of a glycoprotein and amyloid precursor protein (APP)”, APP is a glycoprotein, this sentence’s grammar needs to be corrected
Line 49: “In the normal state, Aβ releases during”, it should be “Aβ is released”
Line 53: “As was established in the culture of neurons”, this should be corrected “As was established in cultured neurons”
Line 57: “its soluble forms are formed that are rapidly released outside the cell and removed”, this needs to be corrected either by removing the word “its” or by replacing the word “that” with “and”
Line 86: “Pls keep microglial health and involves in brain immunity”, it should be “and are involved in”
Line 106: the authors should clarify already here that they are using recombinant Aβ and the type (40 or 42). This is also the case for the methods section 2.3.
Lines 350-351: the authors should explain better the four groups, they do numbering for the first two, but not for the other ones
Comments on the Quality of English LanguageSeveral grammar errors that are mentioned in the comments and suggestions to the authors.
Author Response
I have read with interest the manuscript “Plasmalogens improve clearance of amyloid-beta from mouse brain and cognitive functions” by Shikorov et al. The authors address the potentially beneficial therapeutic effect of plasmalogens against the deposition of amyloid beta. They employ non-transgenic mice with two main series of experiments, either by injecting recombinant amyloid-beta fibrils or by following the endogenous amyloid-beta accumulation. In both cases they treat the mice with plasmalogens and assess the effect by ELISA, immunofluorescence, and behavioural tests.
Overall, the study is interesting and obviously the use of in vivo models is of higher importance and amount of work. However, there are several problems throughout the manuscript that need to be addressed before proceeding for publication.
Comment: First of all, there are many grammatical errors especially in the introduction, but also in other parts of the text. I listed some of them at the end of my comments and I encourage the authors to correct them and carefully proof-read the manuscript.
Response: The authors would like to thank so much the reviewer for the positive assessment of our research and detailed review of the results, as well as for important advices and recommendations for improving the quality and style of our article. We have corrected all grammatical and stylistic inaccuracies and also improved the English language. All changes in the text of the article are highlighted in yellow.
Comments: The mice employed for this study are non-transgenic mice. However, the authors demonstrate a time-dependent accumulation of Aβ42 (Figure 5). Is this a finding that has been previously reported by other groups? I tried to find citations in the manuscript supporting that, but I was not able to locate them. I find this quite confusing, because there are several transgenic mouse models of Aβ deposition in the literature, so I am not sure whether this is a known feature for mice to deposit Aβ42 naturally over time. If they do, what is the homology between the human and murine version? Have the authors addressed biochemically or by stainings the nature of these Aβ42 species? Perhaps detergent-insoluble centrifugation assays or stainings with amyloid dyes could elp.
Response: Figure 5 shows the increase in brain Aβ levels of mice with age. This figure does not show the dynamics of Aβ accumulation in AD mice. Age-related increases in the brain Aβ levels of mice and humans have also been shown in other studies that we cite in our article (Lines 582-608).
Comments: The authors report their systems as AD or injected AD based on injections and IF stainings of Aβ42. I would argue that this is not the case, because to classify human patients as AD cases there needs to be the concurrent deposition of Aβ plaques and Tau neurofibrillary tangles. Additionally, the filamentous structures of these amyloids are very specific and reported in detail over the last years (PMID: 28678775, 35025654). Therefore, the models mentioned here only partially replicate the AD situation and this is something that should be mentioned. Perhaps the authors should mention them as AD-like mouse models, since they do not recapitulate the main aspects of AD pathology.
Response: We fully agree with this fair comment. An explanation of the choice of the injection model of AD for our studies and the need for additional investigations using transgenic models of AD are added to the limitations (Lines 609-649).
Comments: There are also several issues with the presentation of the findings.
- Even though I appreciate the effort of the authors to visualise the design of their experimental approach, the schematic of Figure 1 is overloaded with lots of information and needs to be simplified. The artistic drawings in Figure 1a and b should also be removed, as they are not related to the nature of the experiments. Additionally, what does EIA mean and why the authors do not use the term ELISA? Overall, I think that the schematics of Figure 1g and h are enough to illustrate the experiment.
Response: We have corrected Figure 1 and its legend accordingly (Lines 144-151).
Comments: Figure 2: Multiple issues that must be addressed:
Why did the authors perform NG2 labelling for the brain samples instead of DAPI for the rest of the regions? (Note: the same question applies to Figure 5). The authors also do not report the antibody for the NG2 staining.
Response: NG2 is a pericyte marker. Since the focus of our research was on studying the soluble form of Aβ, i.e., which is capable of moving with the cerebrospinal fluid flow, we used this marker to also analyze the deposition of Aβ in the perivascular spaces that are filled with the cerebrospinal fluid. We have shown this clearly in Figure 2 c and d in the first version of the article. We have added information about antibodies for NG2 to the methods (Line 350).
Comments: The presentation is rather poor. Starting with the brain images, the control top left image seems to have very high levels of Aβ, way higher than the injected samples. Additionally, for the zoomed areas the authors should present images from areas that are similar in all cases. While the control zoomed images are similar, the AD and the AD + Pls are from completely different areas. The authors should improve this, because it is important to display representative images that are in line with the quantified data. The scale bars are all in different places in the panels, there needs to be concistency on that point so that it is more presentable. Additionally, why are the scale bars of the AD samples for the dcLNs different compared to the others?
Response: We have improved Figure 2.
Comments: Figure 5: The dcLNs representative images are very different from the equivalent ones in Figure 2. Why is that and how do the authors comment on it?
Response: The cervical lymph deep nodes shown in Figures 2 and 5 are from different series of experiments and groups. Figure 2 presents data on the effects of Pls on Aβ content in the brain, its meninges, and lymph nodes in 4 groups including control, control+Pls, mice with Alzheimer's disease without and after the Pls therapy. Figure 5 shows 16 age groups receiving and no PLs, i.e., 4-fold more representative images. Therefore, Figure 5 presents confocal images of lymph nodes only at magnification.
Comments: Lines 359-364, 406-413: very confusing comparison for the ELISA results as they describe them in the text, almost impossible to follow. The authors should either plot them in graph or in table. Additionally, the levels of Aβ in the uninjected animals are estimated at about 10 pg/g tissue and after injection it goes up to 23.4 pg/g tissue, while in the 24-month-old uninjected mice the levels are about 18 pg/g tissue. To me this sounds weird, as I would expect the injected animals to have way higher levels of Aβ.
Lines 350-351: the authors should explain better the four groups, they do numbering for the first two, but not for the other ones.
Response: We have moved the ELISA data into Table 1. To create an injection model of AD, 3-month-old mice were used. Their brain Aβ level was 10.18±1.23 pg/g tissue. After the intrahippocampal injection of Aβ, its level increased by 2.3 times (23.43±1.14 pg/g tissue vs. 10.18±1.23 pg/g tissue, p<0.001), while in old mice it increased by 1.8 times (18.83±1.07 tissue vs. 10.18±1.23 pg/g tissue, p<0.001). High Aβ levels in the brains of old animals (24-month-old mice correspond to 80 years of human life: Life Sciences 152 (2016) 244–248) are associated with a dramatic decline in the function of the meningeal lymphatic vessels (Nature 2018, 560(7717), 185-191; Nature 2019, 572(7767), 62-66), which leads to a significant brain accumulation of toxic peptides. We have added explanations in the text for the animal groups to make the results more understandable and transparent (Lines 395-396).
Comments: Specific comments for grammar errors:
Line 45: “which is involved in pathogenesis”, missing the word “the”
Lines 48: “result of the cleavage of a glycoprotein and amyloid precursor protein (APP)”, APP is a glycoprotein, this sentence’s grammar needs to be corrected
Line 49: “In the normal state, Aβ releases during”, it should be “Aβ is released”
Line 53: “As was established in the culture of neurons”, this should be corrected “As was established in cultured neurons”
Line 57: “its soluble forms are formed that are rapidly released outside the cell and removed”, this needs to be corrected either by removing the word “its” or by replacing the word “that” with “and”
Line 86: “Pls keep microglial health and involves in brain immunity”, it should be “and are involved in”
Line 106: the authors should clarify already here that they are using recombinant Aβ and the type (40 or 42). This is also the case for the methods section 2.3.
Response: We have corrected English and grammar errors throughout the article.
The authors would like to express their sincere gratitude for the opportunity to improve our article with constructive suggestions and critical comments.
Authors

Round 2
Reviewer 1 Report
Comments and Suggestions for Authors
I am appreciative to the authors for their reflections on and actions to my suggestions and comments.
Author Response
The authors thank the reviewer for the opportunity to discuss our results and for the great support of our research.

Reviewer 2 Report
Comments and Suggestions for Authors
A great effort has been made in improving the quality of content and presentation of the manuscript. The authors successfully addressed my concerns and questions.I am pleased to recommend the manuscript for publication in IJMS.
Author Response
The authors are grateful to the reviewer for the great help with our article and the opportunity to improve it for possible publication in IJMS.

Reviewer 3 Report
Comments and Suggestions for Authors
I have read the revised version of the manuscript “Plasmalogens improve lymphatic clearance of amyloid-beta from mouse brain and cognitive functions” by Shikorov et al. The authors responded to my comments, they have improved the presentation of their data as requested and corrected grammatical as well as syntactical errors in the text.
However, my main concern remains as described in my report regarding the original version and it has not been addressed adequately by the authors: the authors use non-transgenic mice and report the time-dependent increased levels of amyloid beta 42 in the mouse brains in absence of injections with recombinant amyloid beta 42 or other treatments. The specific presence of amyloid beta 42 is very important, because it is severely more aggregation-prone compared to other amyloid beta species and I haven’t come across any other mouse model in literature with similar phenotype. Therefore, I asked the authors to support this finding with appropriate citations and they directed me in their response to the following 5 papers as cited in the manuscript:
Citation 24: this paper reports the time-dependent accumulation of endogenous amyloid beta 40 (but not amyloid beta 42) in the cell culture supernatant from rat primary neurons and in Yorkshire pigs, so no mice involved
Citation 27: no reporting on amyloid beta 42 from non-transgenic mice, but they use several transgenic mouse models
Citation 28: no mentioning of amyloid beta 42
Citations 30 and 31: both are review papers discussing the processing of APP and the amyloid beta in ageing for humans
Therefore, the authors describe an unprecedented finding: the time-dependent increase of amyloid beta 42 levels from non-transgenic mice, without providing any characterisation data towards those species and with experimental approaches that are prone to errors. For example, according to the methods, the mice are not perfused before proceeding with immunohistochemistry analysis, which is very problematic since the antibody against amyloid beta 42 is generated in mice and this would massively increase the non-specific detection. Another example is the ELISA and the immunohistochemistry methods that are employed and they would both certainly detect the injected material of recombinant amyloid beta 42 species. Most importantly, if the generation of amyloid beta 42 is so effective from non-transgenic mice, why do they perform injection experiments with recombinant insoluble amyloid beta 42 filaments and why are there so many transgenic mouse models for amyloid beta in literature? Especially regarding the injection experiments, this does not align with the authors’ argument that they are addressing “…the Pls effects on the lymphatic removal of soluble forms of Aβ from the brain” (line 635-636). If they are studying the clearance of soluble amyloid beta, why bother injecting stereotactically in the brain recombinant amyloid beta 42 aggregated species?
Overall, I must admit that I find the study interesting and I have a lot of respect towards the use of animals in these experiments. Therefore, I think it is essential to justify why these animals were used and I would like to give the authors the opportunity to respond and provide more experimental data and/or proper citations that could support the endogenous amyloid beta 42 increase in the mouse models and the appropriate detection from their methods.
Comments on the Quality of English Language
English is fine, much improved compared to the original version.
Author Response
The authors highly appreciate the reviewer's comments and would like to express our gratitude once again for the opportunity to discuss our results. We also apologize for the misunderstanding. We responded to the same comment last time that Figure 5 shows data from mice of different ages. This is not an injection model of Alzheimer's disease.
In our studies, we used two functional models: an injection model of Alzheimer's disease and age-related changes in the brain.
An increase in brain Aβ (42) levels was shown only in the second model, i.e. with age. An age-related increase in Aβ in the brain of healthy mice and rats has been shown in other studies (Journal of Alzheimer’s Disease 95 (2023) 719–733; J Alzheimers Dis 2018, 61(4), 1425-1450) as well as in ours (J Vis Exp. 2024 28;(208). doi: 10.3791/67035), i.e. the fact of age-related increase in brain Aβ (42) levels is well known. We discuss about it (Lines 582-608).
We published a method for detecting brain Aβ (42) levels in mouse of different ages earlier (J Vis Exp. 2024 28;(208). doi: 10.3791/67035). In this our study, we used the same protocol.
We improved citations in the discussion (Lines 582-648).
The authors are grateful for the detailed review of our article and the opportunity to improve it with the help of the reviewer's constructive recommendations.
Authors

Round 3
Reviewer 3 Report
Comments and Suggestions for Authors
The authors addressed my concerns by providing additional references and therefore, I endorse its publication.